# Oxidative-Stress Related Gene Polymorphism in Endometriosis-Associated Infertility

**DOI:** 10.3390/medicina58081105

**Published:** 2022-08-15

**Authors:** Traian Irimia, Lucian Pușcașiu, Melinda-Ildiko Mitranovici, Andrada Crișan, Mihaela Alexandra Budianu, Claudia Bănescu, Diana Maria Chiorean, Raluca Niculescu, Adrian-Horațiu Sabău, Iuliu-Gabriel Cocuz, Ovidiu Simion Cotoi

**Affiliations:** 1PhD School of Medicine, University of Medicine, Pharmacy, Science and Technology ”George Emil Palade” Targu Mures, 540139 Targu Mures, Romania; 2Department of Obstetrics and Gynecology, Emergency County Hospital Hunedoara, 331057 Hunedoara, Romania; 3Department of Pathology, County Clinical Hospital of Targu Mures, 540072 Targu Mures, Romania; 4Department of Pathophysiology, George Emil Palade University of Medicine, Pharmacy, Science and Technology of Targu Mures, 540142 Targu Mures, Romania

**Keywords:** endometriosis, infertility, oxidative stress

## Abstract

*Background and Objectives:* Endometriosis is a benign inflammatory disease associated with infertility and chronic pelvic pain, estimated to affect 7–10% of reproductive-age women, with the possibility of malignant transformation. Recent studies focus on oxidative stress and genetic mutations as risk factors in the pathophysiology of endometriosis-associated infertility. *Materials and Methods:* This case-control study is the first in Eastern European women that aimed to investigate four genes’ genetic polymorphisms that encode antioxidant enzymes involved in oxidative stress (glutathione peroxidase 1, GPX1 198Pro > Leu, catalase CAT-262C > T, glutathione S-transferase M1, and T1 null genotype) and their association with endometriosis-related infertility. We compared 103 patients with endometriosis-associated infertility with 102 post-partum women as the control group. *Results:* The endometriosis group had a mean age of 34.5 +/− 6.12 years, while the control group’s mean age was 35.03 +/− 5.95 years. For CAT-262C > T polymorphism, the variant genotypes were significantly more frequent in the endometriosis group. Moreover, for the GPX1 198Pro > Leu, the endometriosis group had significantly more frequent CT and TT genotypes. The null genotype of GSTM1 was detected significantly higher in the endometriosis group. No significant differences were found in the frequency of GSTT1 between the two groups. This study suggests that GPX1 198Pro > Leu, CAT-262C > T, and GSTM1 polymorphisms may be risk factors and that the association between the GSTM1-GSTT1 null genotype may play a significant role in endometriosis-associated infertility. Moreover, this study suggests that the GSTT1 null genotype does not influence the disease. Visual identification of endometriotic lesions with microscopic confirmation is the accepted gold standard for diagnosing endometriosis, but general anesthesia and laparoscopy are required. *Conclusions:* In this regard, a panel of genetic or laboratory markers is needed for the early diagnostics of this prevalent disease, especially in the case of young patients with future pregnancy intention.

## 1. Introduction

Endometriosis is a benign inflammatory estrogen-dependent disease associated with infertility and chronic pelvic pain, estimated to affect 7–10% of women of reproductive age [1,2,3]. Ectopic endometrial glands and stroma with a yet undetermined etiology adhere to different sites, leading to various signs and symptoms [4]. The peritoneal fluid contains pro-inflammatory cytokines, angiogenic and chemotactic factors, and low antioxidant status [3].

The etiology of the disease remains unclear, but the influence of environmental and genetic factors has been determined in several studies [2,5,6,7,8,9,10,11,12]. Different genetic factors have been identified as being risk factors for developing the disease [2]. Moreover, DNA variation influence is being researched as having a major role in the pathogenesis of this disease.

While single nucleotide polymorphisms have no functional consequence, their combination predisposes women to the disease’s inception.

The involvement of inflammatory processes in endometriosis is a widely accepted phenomenon [4,13,14]. The endometriosis lesions invade the adjacent tissues and proliferate by developing their vascularization. This development is influenced by cytokines and growth factors such as interleukins 1, 8, tumor necrosis factor, transforming growth factor, and interferon-gamma [4]. Recent studies focus on oxidative stress and genetic variants as risk factors in the pathophysiology of endometriosis [1,2,3,4,14]. An imbalance between reactive oxygen species (ROS) (increased levels) and antioxidants (decreased levels) may lead to the disease and subsequent infertility. Oxidative stress has essential roles in metabolic and biological systems [1,2,3,4,14].

Elevated levels of ROS affect mitochondria within oocytes and embryos, and may stop cell division and promote cellular injury [3]. Moreover, an increase in nitric oxide (vasodilatory properties) leads to altered vascular function, potentially resulting in infertility [3].

Levels of ROS are balanced with antioxidants, neutralizing the destructive effect on cell structures. The antioxidants can be enzymatic (superoxide dismutase, catalase, glutathione peroxidase, glutathione oxidase) and non-enzymatic (GSH, i.e., glutathione, is used by GPX, glutathione peroxidase) [3].

GPX1 (glutathione peroxidase 1) and CAT (catalase) are antioxidant enzymes that control hydrogen peroxide concentrations. The GSH family includes GPX, GST (glutathione S-transferases), and glutathione reductase [5,6]. GPX1, a cytosolic selenium-dependent enzyme, has a functional nucleotide polymorphism located at codon 198, which substitutes proline with leucine. This alteration causes the Leu variant to be a less effective antioxidant enzyme [5]. Catalase has a functional nucleotide polymorphism consisting of cytosine (C) substitution with thymine (T) at position 262. Compared to the C allele, the T allele has higher transcriptional activity, increasing gene expression levels. However, the CAT −262C > T polymorphism decreases the catalase activity and thus influences the reaction to oxidative stress [6].

The GST (glutathione S-transferases) enzyme family has a role in the metabolism of different carcinogens, drugs, and toxins. GSTM1 and GSTT1 are the most extensively studied classes due to their frequent allelic variants. There are controversies on the association between polymorphisms of GST gene family and endometriosis [7,15,16,17,18,19,20,21,22,23]. GSTM1 and GSTT1 are believed to have an essential role in detoxifying the end-products of oxidative stress released due to the repairing process of the ovarian epithelium post ovulation. Null alleles of GSTM1 and GSTT1 lead to inadequate detoxification and thus to increased levels of inflammation [7,8,22,23,24,25].

The development of oocytes and ovulation is in close relationship with ROS and antioxidants, with their imbalance altering this process and causing infertility, polycystic ovary syndrome, and endometriosis. Moreover, if pregnancy does occur, increased ROS may lead to complications involving the placenta, abortion, recurrent pregnancy loss, intrauterine growth restriction, preeclampsia, and preterm labor [3].

The aim of this study was to investigate four genetic polymorphisms of the antioxidant enzymes involved in oxidative stress (glutathione peroxidase Pro198Leu, catalase C−262T, glutathione peroxidase S-transferase M1 null allele, and glutathione peroxidase S-transferase T1 null allele) and their association with endometriosis-related infertility.

## 2. Materials and Methods

We prospectively assessed 103 patients with endometriosis-associated infertility. All subjects underwent laparoscopic surgery at one academic hospital in Romania between 2015–2019.

The inclusion criteria were: infertile females aged 24 to 40 years, with BMI ranging from 19 to 27, with visual and microscopic confirmation of endometriosis. We excluded patients with chronic treatments that may interfere with fertility, and we also ruled out women whose partners were not tested for fertility disorders.

We selected 102 consecutive post-partum women with no history of obstetrical diseases and no fertility treatment before pregnancy for the control group. All the participants signed an informed consent form.

Surgery with a laparoscopy approach was performed for the endometriosis group (*n* = 103) to confirm the diagnosis and treat the lesions. The indications were either symptoms or infertility. The endometriotic lesions were treated with ablation for peritoneal lesions, cystectomy, or cyst drainage for ovarian endometrioma. In order to exclude other factors causing infertility, tubal chromopertubation was routinely performed perioperatively. Blood samples were taken on days 2–4 postoperatively. For the control group (*n* = 102), blood samples were collected day 2–4 post-partum, after vaginal or cesarean delivery, during the same period.

Genotyping of glutathione peroxidase (GPX) Pro198Leu and catalase (CAT)-262C > T, glutathione S-transferases (GST) GSTM1, GSTT1 gene polymorphisms were performed by using genomic DNA (gDNA). Genomic DNA of the patients and controls was extracted from peripheral blood collected in EDTA tubes using Genomic DNA Mini Kit (Thermo Fisher Scientific, Norcross, GA, USA)*,* following the manufacturer’s recommendations.

GPX1 198Pro > Leu and CAT-262C > T were genotyped by applying the PCR-RFLP (polymerase chain reaction and restriction fragment length polymorphism) technique. In contrast, the multiplex PCR method established GSTM1 and GSTT1 null genotypes, as previously described.

The statistical analysis was performed using STATA version 16.1. Descriptive statistics assessed the similarity between the two groups. The chi-square test was used to evaluate the genotype distribution and frequencies between the two groups having discrete data sets. When necessary, Yates’s and Fisher’s corrections were applied. Alpha was set at 0.05. Odds ratios (ORs) and confidence intervals (95% CI) were calculated to assess the risk of developing the disease for each genotype.

The Ethics Committee of the University of Medicine, Pharmacy, Science and Technology, “George Emil Palade” Targu Mures, Romania, approved the study (No 6253/4 March 2022).

## 3. Results

The endometriosis-associated infertility group had a mean age of 34.5 +/− 6.12 years, while the mean age for the control group was 35.03 +/− 5.95 years. The rAFS (revised American Fertility Score) distribution showed a predominance of stage III of the disease with 72.79% cases, while stages II and IV represented 26.3% cases.

The genotype frequencies for GPX1 198Pro > Leu (198C > T) and CAT-262C > T among patients with endometriosis and controls (wild-type homozygous genotype CC, heterozygous genotype CT, and variant homozygous genotype TT) are shown in Table 1.

The CAT-262C > T variant homozygous genotype (TT) and heterozygous genotype (CT) represent risk factors in endometriosis development (*p* = 0.013 for CT and *p* = 0.019 for TT). Moreover, we found significant associations between variant genotypes (CT, TT) of GPX1 198Pro > Leu and the risk of developing endometriosis (*p* = 0.040 for CT and *p* = 0.019 for TT).

The null genotype of GSTM1 was significantly higher in the endometriosis group (*p* < 0.0001). No significant differences were found in the frequency of GSTT1 between the two groups.

The other investigated genotypes did not show an association between gene polymorphisms and the risk of developing endometriosis.

## 4. Discussion

Both groups were selected to have a similar number of observations as well as age distribution and no previous medical diagnoses that might interfere with that of endometriosis-associated infertility, thus reducing possible unforeseen biases.

The literature’s interest has shifted in the last decade and there has been more research on the involvement of a genetic predisposition in the development of certain diseases.

Several studies [1,2,3,4] have assessed the implications of reactive oxygen species and antioxidants, trying to find a link between different risk factors and endometriosis-associated infertility. Most authors agree that endometriosis is a chronic inflammatory disease that can cause infertility and recurrent miscarriages [1,2,3,4,15]. Still, the exact species of ROS and antioxidants whose balance is affected must be determined. ROS are involved in processes such as folliculogenesis, maturation of the oocyte, the function of the corpus luteum, embryogenesis, and placental development [1,2,3,4,15]. The imbalance between ROS and antioxidants is a known cause of fertility disorders [1,16,17]. The inflammatory status increases according to the progression of endometriosis, thus decreasing the chances that a spontaneous pregnancy will occur and be carried to term. Researchers have observed a delay in diagnosing endometrial lesions of over 6.7 years [21], despite the poor quality of life. The majority of patients were attributed to rAFS stage III or IV of endometriosis, which might explain the poor reproductive outcome. Thus, the need for an efficient method of predictive diagnosis arises.

The literature is controversial: GPX is an enzyme found in the cytoplasm and mitochondria, eliminating the hydroxyl radical protecting against intracellular DNA damage [11]. Catalase is an antioxidant enzyme that converts hydrogen peroxide, thus saving cells against destruction [10,16]. Several gene polymorphisms of these genes were described. GPX1 198Pro > Leu and CAT-262C > T are most frequently associated with female infertility.

Our study suggested that polymorphisms of GPX1 198Pro > Leu and CAT-262C > T are both associated with the incidence of symptomatic endometriosis. Very few articles address the association between aberrant expression of GPX1 and CAT genes and endometriosis [9,10,12,16,19]. Authors suggest that the implication of GPX1 and CAT gene polymorphisms can cause endometriosis but request more extensive studies to confirm their findings [9,10,12,16,19,23].

According to our results, CAT-262C > T is a risk factor for endometriosis development. The relatively small number of patients is a limitation of this study, and therefore the results must be interpreted cautiously.

GST is a family of antioxidant enzymes, divided into eight classes. Due to the frequent allelic changes, GSTT1 and GSTM1 are the most intensively studied variants [7,20,21,23]. The meta-analysis by Sun-Wei Guo [15] suggests that only GSTT1 is associated with developing endometriosis. Almost a decade later, two other meta-analysis studies [7,21] conclude that both GSTT1 and GSTM1 null genotypes could be risk factors for endometriosis but suggest that further studies are needed for confirmation. In comparison, our findings indicate that only GSTM1 is correlated with endometriosis.

Two meta-analyses [7,21] also concluded that the association of both null genotypes for GSTT1-GSTM1 is related to endometriosis. Our study confirms this theory, as our findings show a powerful association in this case (*p* = 0.0004).

Our results also reveal that GPX1 198Pro > Leu, CAT-262C > T, and GSTM1 null genotype may be risk factors. The association between GSTM1-GSTT1 null genotype may play a significant role in endometriosis-associated infertility. However, GSTT1 null genotype does not influence the disease. [24,25,26,27,28,29,30].

Compared to the existing literature, this discrepancy in our findings might be explained by demographic differences, as ethnicity and environmental factors play an important role in developing the disease.

In addition, in the literature, there are certain immune diseases found together with endometriosis [31]. The immune system is possibly involved in the pathomechanisms of endometriosis. Several studies indicate altered humoral and cellular immunity [23,31,32,33,34,35,36]. Disorders of the immune system accompany every stage of the disease. As a consequence, there are an increased number of macrophages with a reduced ability of phagocytosis, but with an increased ability to secrete pro-inflammatory cytokines, such as IL 6, TNF alfa, IL1 beta, IL8 and prostaglandine [31]. Signaling pathways are also involved, as well as protein interactions with cytokines and cytokine–cytokine receptor interactions, involving interleukynes and drug methabolism cytochrome P450 (involving GSTM1). The differentially expressed genes and the signaling pathways are likely associated with the process of endometriosis [35] and we agree with that in our study.

This is the first study investigating the association of CAT-262C > T, GPX1 198Pro > Leu, GSTT1, and GSTM1 gene polymorphisms in the Eastern European women population with and without endometriosis, having analyzed a large number of patients. However, for a more accurate image, the implication of GPX1, CAT, and GST enzyme activity should be taken into account.

The need to understand the implications of endometriosis-associated infertility on both the patient and her healthcare providers is immense. The focus should be aimed at preventive medicine and early diagnosis, thus having the possibility of treating the cause rather than the effect [24,25,26,27,28,29,30].

The limits of the study consist of how the polymorphisms of GPXI 198Pro > Leu and CAT-262C > T are involved in the malignant transformation, or the GST antioxidant enzymes.

The clinical benefits of our study were that demonstrating the involvement of oxidative stress in the mechanism of endometriosis offers clinical benefits, starting with the modification of diet and drugs involved in its reduction. The consumption of vegetables and fresh fruits is considered beneficial because they contain antioxidants. They also play a proper role in the functioning of the immune system. This seems to play a role in removal of free radicals as well. There is still an urgent need to develop new drugs to modulate oxidative processes and to cure the disease [31,36].

In addition, GSTM1 polymorphism plays an important role in infertility [33], for which reason these patients require treatment adjustment.

By regulating GSTM4 expression we could inhibit proliferation and induce apoptosis [34].

Lastly, it would be necessary to identify the polymorphism that might be of value in screening and early non-invasive diagnosis.

Finally, our research aimed to expand the knowledge base, which eventually yields useful, patient-centered applications. In this regard, endometriosis-associated infertility research should ultimately improve reproductive outcomes, particularly live births, and the utility of early diagnostic and fertility treatments [18].

## 5. Conclusions

Visual identification of endometriotic lesions with microscopic confirmation remains the accepted gold standard for endometriosis diagnosis, but general anesthesia and laparoscopy are required. In this regard, a panel of genetic or laboratory markers is needed for the early diagnosis of this prevalent disease, especially in the case of young patients with future pregnancy intentions. Polymorphism of GPX1 198Pro > Leu and CAT-262C > T are both associated with symptomatic endometriosis, and CAT-262C > T and GSTM1 are risk factors for disease’s development.

## Figures and Tables

**Table 1 medicina-58-01105-t001:** Genotype and allele distributions.

		Cases				Control				*p* Value	OR (95% CI)	OR Interval
		No.	%	No.	%	No.	%	No.	%			
CAT C-262T	CC vs. CT	28	35	52	65	47	54	40	46	0.0135	2.182	1148 to 4086
	CC vs. TT	28	54.9	23	45.1	47	75.8	15	24.2	0.0192	2574	1144 to 5679
	CC vs. CT + TT	28	27.2	75	72.8	47	45.6	55	53.4	0.0050	2289	1253 to 4167
GPX1 Pro198Leu	CC vs. CT	23	25.8	66	74.2	38	40	57	60	0.0415	1913	1001 to 3646
	CC vs. TT	23	62.2	14	37.8	38	84.4	7	15.6	0.0407	3304	1142 to 8537
	CC vs. CT + TT	23	22.3	80	77.7	38	37.3	64	62.7	0.0194	2065	1113 to 3829
GSTM1	Present vs. Null	63	61.2	40	38.8	92	90.1	10	9.8	<0.0001	5.841	2.804 to 11.90
GSTT1	Present vs. Null	82	79.6	21	20.4	82	80.4	20	19.6	0.8889	1.050	0.5315 to 2.091
GSTM1 + GSTT1	Present vs. Null	55	80.9	13	19.1	73	98.6	1	1.4	0.0004	17.25	2.555 to 186.3

CAT C—catalase C, GPX1—glutathione peroxidase 1, GSTM1—glutathione S-transferase M1, GSTT1—glutathione S-transferase T1.

## Data Availability

All data produced here are available and can be produced upon request.

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
