# Peer review of "Oxidative-Stress Related Gene Polymorphism in Endometriosis-Associated Infertility"

_medicina, 2022, doi:10.3390/medicina58081105_

Round 1

Reviewer 1 Report

This is an article well designed and on a very important aspect of endometriosis.

The authors present in a comprehensive way the material and methods of the study. A minor detail that could be added, as the authors selected infertility cases, would be to present the infertility causes.  Also, if the endometriosis of the patients was associated with family history.

Overall, a valuable article that is worth publishing. If the authors consider adding the previously mentioned details, that could expand the research in further associations.

Author Response

Cover letter:

1)Infertility causes were presented line 104-106:             The inclusion criteria were: infertile females aged 24 to 40 years, with BMI ranging from 19 to 27, with visual and microscopic confirmation of endometriosis. We have excluded patients with chronic treatments that may interfere with fertility, and we have also counted out women whose partner was not tested for fertility disorders. So the only one criteria of inclusion was endometriosis with no other obvious cause.

2).Associated endometriosis with family history:In  7% of women endometriosis is associated with their genetic predisposition [31], a small procent. That is why we did not concentrate on this aspect because it is not relevant for clinical benefit.And family history was difficult to trace.

Thank you

Reviewer 2 Report

The paper is an interesting one, with perspectives to be applied in current medical practice.Some observations:

-           I think the investigation and diagnosis protocol should be briefly presented as well as the indication for surgery from the studio batch.

-           It must, beyond the chosen theme for the study, namely oxidative stress, presented in and others (inflammatory molecules such as cytokines -interleukin-1β, interleukin-6, tumor necrosis factor (TNF), monocyte chemoattractant protein 1, etc , prostaglandins, and metalloproteinases).

-           There are also a number of related studies theory of immunological dysfunction, several research studies focused on the role of humoral and cellular immunity in the generation and progression of the disease.

-           It is important from the point of view of the pathological consequences that imbalance between Reactive Oxygen Species - ROS (increased levels) and antioxidants (decreased levels) may develop the disease and subsequent infertility.

-           It is natural to ask what mechanisms are activated, what are the links through which we pass. An opinion would be welcome, as a consequence of the work done.

-           What is the clinical benefit of the study in current practice?

-           The bibliographic title must be adjusted, the whole collective must not be mentioned in the variant (example et al).

-           Bibliographic data for arguments and discussions regarding GSTM1 should be updated. Obviously with impact on resident material. There have been enough studies since 2020-2021.

Author Response

 Cover letter:

1).Investigation an diagnosis protocol briefly presented from line 119-126, with the genotipic tests.The indication for surgery:I presented it line 111-114, now I added only the reason for surgery.

2).Other inflammatory molecules:Implantation in the peritoneum of endometrial cells has as cosequence an increased production of cytokine including tumor necrosis factor alfa (TNF alfa)and interleukines[31].It is also  emphasized from line 56-59.

3). Immunological disfunction , humoral and cellular immunity in the generation and progression of the disease:There are in literature data on certain immunedisease found toghether with endometriosis[31]. Immune system is implicated in the possible pathomechanism of endometriosis. Several studies indicate altered humoral and cellular immunity.[31-33].Disorder of the immune system accompany every stages of the disease. As consequence there are an increased number of macrophages withe a reduce ability of phagocytosis but with an increased ability to secrete pro-inflammatory cytokines:IL 6, TNF alfa, IL 1 beta, IL8 and prostaglandine.[31].Line 214-220.

4). What mechanism are activated, links through which we pass: Signaling pathways were involved, protein interaction with cytokines, cytokine-cytokine receptor interaction, involving Interleukynes, drug methabolism cytochrome P450 (involving GSTM1). The differentially expressed genes and signaling pathways are likely associated with the process of endometriosis[35] and we agree with this in our study.Line 220-224

5).Clinical benefit in current practice:Demonstrating the involvement of oxidative stress in the mechanism of endometriosis offers clinical benefits starting withe the modification of the diet up tu drugs involved in its reduction. The consumption of vegetables and fresh fruits is considered beneficial because they contein antioxidants. They play proper role in the functioning of immune system. It seems to play a role in removal of free radicals. There is still an urgent need to develope new drugs to modulate oxidative process and cure the disease.[31,36].

In addition GSTM1 polymorphism plays an important role in infertility[33], reason for which these patients requires treatment adjustment.

By regulating GSTM4 expression we could inhibit proliferation and induce apoptosis [34].

And last but not least it would be necessary to identify the polymorphism that might be of value in screening.Line 238-250

6).In bibliographic data  we could adjuste in order the whole collectve not to be mentioned, but the templates give me the exact manuscript, as I used here. And unfortunately I had  an unpleasant experience with a co-author not mentioned in the bibliography.Thank you for understanding.

7).I updated the bibliographic data as you can see in discussion.Thank you
